# Hippocampal Subregion and Gene Detection in Alzheimer’s Disease Based on Genetic Clustering Random Forest

**DOI:** 10.3390/genes12050683

**Published:** 2021-05-01

**Authors:** Jin Li, Wenjie Liu, Luolong Cao, Haoran Luo, Siwen Xu, Peihua Bao, Xianglian Meng, Hong Liang, Shiaofen Fang

**Affiliations:** 1College of Intelligent Systems Science and Engineering, Harbin Engineering University, Harbin 150001, China; lijin@hrbeu.edu.cn (J.L.); liuwenjie@hrbeu.edu.cn (W.L); caoluolong@hrbeu.edu.cn (L.C.); 13945056105@163.com (H.L.); xusiwen@hrbeu.edu.cn (S.X.); baopeihua@hrbeu.edu.cn (P.B.); 2School of Computer Information and Engineering, Changzhou Institute of Technology, Changzhou 213032, China; mengxl@czust.edu.cn; 3Department of Computer and Information Science, Indiana University-Purdue, University Indianapolis, Indianapolis, IN 46202, USA

**Keywords:** hippocampus, random forest, subregion, genetic algorithm, clustering evolution

## Abstract

The distinguishable subregions that compose the hippocampus are differently involved in functions associated with Alzheimer’s disease (AD). Thus, the identification of hippocampal subregions and genes that classify AD and healthy control (HC) groups with high accuracy is meaningful. In this study, by jointly analyzing the multimodal data, we propose a novel method to construct fusion features and a classification method based on the random forest for identifying the important features. Specifically, we construct the fusion features using the gene sequence and subregions correlation to reduce the diversity in same group. Moreover, samples and features are selected randomly to construct a random forest, and genetic algorithm and clustering evolutionary are used to amplify the difference in initial decision trees and evolve the trees. The features in resulting decision trees that reach the peak classification are the important “subregion gene pairs”. The findings verify that our method outperforms well in classification performance and generalization. Particularly, we identified some significant subregions and genes, such as hippocampus amygdala transition area (HATA), fimbria, parasubiculum and genes included *RYR3* and *PRKCE*. These discoveries provide some new candidate genes for AD and demonstrate the contribution of hippocampal subregions and genes to AD.

## 1. Introduction

With recent technological advances of imaging genomics studies, a large amount of imaging data and genetic data have been collected on the human brain. These data provide an unprecedented opportunity to examine the effects of genetic variation on the brain. Based on these data, research on neuroimaging makes it possible to detect brain changes in AD patients. The genome-wide association study (GWAS) [1] is used to analyze the association between single nucleotide polymorphism (SNP) and pathological phenotypes. Therefore, the fusion of imaging and genetic data may provide a new insight for AD research.

Hippocampus is a combination of subregions with different functions [2,3,4,5], and the study of subregions furthers the understanding of the hippocampal mechanism. For example, the volumes of cornu ammonis (CA) 3 region and CA4 were decreased in major depression patients [6] and shrinking of the molecular layer of the dentate gyrus (DG-ML) volumes were related to delayed memory [7]. The parasubiculum involved in the connection between the hippocampus and the cortex subcortical areas, and was responsible for memory [8,9]. Thus, subregions selected as phenotypes were worthy for further research.

In the past decade, structural magnetic resonance imaging (MRI) and functional MRI have been used in mild cognitive impairment (MCI) research. For example, a decrease in the volume of gray matter in the middle temporal lobe was detected in MCI subjects [10]. Another functional brain network study showed that the shortest path length of MCI subjects was greater than that of HC group [11]. The combination of single indicator integrated the different information between them, which was superior to the classification performance of single one. For example, Wee et al. constructed a brain network based on structural MRI and functional MRI data and extracted local clustering coefficients from the brain network to perform MCI recognition [12]. The MCI participants were divided into two groups (early MCI and late MCI) according to the severity of amnestic impairment in ANDI. Among these participants, the early MCI (EMCI) group met the following criteria: 1 standard deviation ≤ memory test performance - standardized norms ≤ 1.5 standard deviation. The late MCI (LMCI) group met the following criteria: memory test performance - standardized norms ≥ 1.5 standard deviation. In the research of Tripathi et al., the voxel-based features and imaging structure were applied to classify the EMCI and LMCI [13]. In recent research, an interesting method to construct the fusion feature using imaging data and gene sequences was described in [14]. In addition, the correlations such as heritability and *p*-value between AD group and HC group are quite different. This may bring a new sight for indicator combination.

However, the classic analysis methods did not perform well in classifying fusion features [15,16,17]. In a recent research, Zheng et al. proposed a selection method based on sparse linear regression [18]. Another method that combined clustering and bee colony algorithm was used to solve the problem of multidimensional data [19]. A clustering evolutionary random forest described in [14] was applied to predict the group of samples and discovered the important “brain region-gene pairs”. However, it is still challenging to detect the fusion features constructed by the correlations and genes.

Drawing on the correlations and the ideas of the above research, we proposed a novel link between hippocampal subregions and genetic data using the correlations and genes to reduce the diversity in same group. To classify the sample labels and find important features, we proposed the genetic clustering random forest method based on the genetic algorithm. We firstly calculated the fusion features using correlations and genes to amplify the difference between AD and HC group. Then we used the genetic clustering random forest method based on genetic algorithm for model construction and model training. Subsequently, we applied the best parameter combinations to extract the important features from the test set and calculate the classification accuracies. Finally, we used EMCI and LMCI datasets to evaluate the generalization of our method. The experiment results demonstrate that the identified abnormal subregions and pathogenic genes will further our understanding of the underlying mechanisms of AD.

## 2. Materials and Methods

### 2.1. Imaging and Genotype Data

In total, we downloaded 387 samples with imaging and genotype data from ADNI (adni.loni.usc.edu), including 262 HC and 124 AD subjects (We have obtained permission to use data from ADNI, and the approval date is October 7, 2020.). We analyzed the HC and AD groups with the genetic data and the MRI scans separately. Details of participants’ information are shown in Table 1.

MRI scans were preprocessed using voxel-based morphometry (VBM) and then segmented and normalized to the Montreal Neurological Institute (MNI) space. An 8 mm FWHM (full width at half maxima) kernel was applied to the segmented and extracted gray matter density (GMD) maps for smoothing. The automatic anatomical labeling (AAL) atlas [20] was employed to define the regions of interest and their coordinates (left hippocampus and right hippocampus).

We used the process described in [21,22] to select SNPs. Briefly, according to the manufacturer’s protocol, all ADNI participants were genotyped using Illumina GWAS arrays (610-Quad, OmniExpress or HumanOmni2.5-4v1) (Illumina, Inc., San Diego, CA, USA) and blood genomic DNA samples [23,24]. Then quality control was performed for the SNPs obtained from ADNI using PLINK v1.9 [25]. SNPs meeting all the following criteria were extracted: (1) SNPs on chromosome 1–22; (2) call rate of each SNP was above 95%; (3) minor allele frequency was above 5%; (4) Hardy–Weinberg equilibrium test *p* was above 1.0 × 10^−6^ and (5) call rate of each participant was above 95% [21,22]. Overall, 563,980 SNPs that passed the QC were included in the following analyses.

We performed GWAS using the image data and genetic data in the hippocampus using the linear regression in PLINK. Age, gender, education and the top 10 principal components from population stratification analysis were included as covariates. Finally, Bonferroni correction was performed on the GWAS results to control for multiple comparisons.

The Manhattan plots of CA1 of HC and AD are shown in Figure 1 [26].

### 2.2. Construction of Fusion Features

To detect the correlation between hippocampal subregions and genes, we firstly constructed the fusion features of subregions and genes. Each SNP corresponded to a base (A, T, C, G), and each gene contained multiple SNPs. If the base was recoded by a number, then the gene was regarded as a set of multiple numbers. This combination of number was defined as a gene sequence. In the linear regression, the direction of the regression coefficient represents the effect of each extra minor allele (A1) (i.e., a positive regression coefficient means that the minor allele increases risk/phenotype mean). Since we used linear regression for GWAS, we chose the minor allele for the corresponding gene number sequence (for example, if a gene is “AACGGTCA”, the corresponding gene sequence is “[1, 1, 3, 4, 4, 2, 3, 1]”). In the AD group, we found that the variances and correlations of the hippocampal subregions explained by SNPs were quite different than in HC group, and the SNPs with little changes had little or no contribution to AD. Using these correlations and gene sequence to construct fusion features, the differences were further amplified between the AD and HC groups, making it easier to detect related genes and regions.

Firstly, the hippocampus of resulting images was segmented into 12 subregions [2] (Figure 2) and combined with genetic data for genome-wide association studies. The results represented the correlation between subregions and SNPs were kept, such as heritability, regression coefficient and asymptotic *p*-value. Secondly, we used GATES (gene-based association test using the extended Simes procedure) and Genome Reference Consortium Human build 37 (also known as “hg19”) [27,28] to map 563,980 SNPs onto 24,894 genes according to their based positions and the chromosome they belong. Among these genes, the largest number of SNPs is 1415, and the smallest number is one. Thirdly, we selected genes based on the number of SNPs they contained. Among them, genes with SNPs number ≥ *N**snp* were defined as top *N**gens* genes. Then, the digital sequences of genes were obtained by recoding the four bases into digits (A -> 1, T -> 2, C -> 3, G -> 4). For the *N**snp* SNPs in one gene, the set of the corresponding *N**snp* correlations (such as the corresponding *N**snp* heritability) was defined as a correlation sequence. Furthermore, the correlation sequences and gene sequences were adjusted into several groups according to SNP numbers. As the optimal method that was described in [14], the Pearson correlation analysis was introduced to construct the “subregion-gene pairs”.

### 2.3. Construction of Genetic Clustering Random Forest

The multimodal data research was faced with the challenge of large capacity and multiple styles. As a representative algorithm of ensemble learning, random forest had desirable processing capabilities for such data. Therefore, the genetic clustering random forest method was performed in this paper. The random forest and genetic algorithm were combined to evolve decision trees genetically. Through hierarchical clustering of the resulting trees, the features that classified AD and HC better were gradually selected from the original dataset. The schematic diagram of genetic clustering random forest is described in Figure 3.

The original sample set *S* is defined as
(1)$$S=\{x_{i},y_{i}\},i\in [1,N]$$
where *x*_*i*_ donate the features in data set, and *y*_*i*_ = {−1, 1} donate the corresponding label of *x*_*i*_. (HC = 1, and AD = −1). *N* is the total number of features.

The training set *S*_*tra**i**n*_, validation set *S*_*v*_ and test set *S**test* are extracted according to *S*. Additionally, the ratio of *S*_*tra**i**n*_:*S*_*v*_:*S**test* is 5:3:2. Then, *f**i**x* (*N**gens* × 12) features and labels are randomly selected from *S*_*tra**i**n*_. The *f**i**x*(*x*) is the rounding function, the *N**gens* is the number of selected genes and 12 is the number of hippocampal subregions. Finally, we used the selected features and labels to construct the decision trees.

To obtain the initial random forest, *n* decision trees were constructed by repeating the method above for *n* times.

The Euclidean distance was introduced to detect the similarities between decision trees in the random forest. The formula was defined as
(2)$$d_{e}=\sqrt{\sum_{i=1}^{n}{\left(x_{1i}-x_{2i}\right)}^{2}}$$
where *d_e_* is the Euclidean distance. *x_1i_* and *x_2i_* are the features in two decision trees.

The decision trees in random forest were taken as the initial population, and 2 groups of 5 trees were chosen randomly. For each group, the similarities between trees were calculated using Equation (2), and the tree pair with the biggest similarity was extracted as the candidate parent. Among the four candidate parents, the group with the closest similarity was regarded as a parent group, and a new decision tree was then generated. Another tree was generated by the group having the second-ranked similarity. The schematic diagram of genetic evolution is described in Figure 4.

A new random forest was constructed by repeating the step above for *n*/2 times.

The similarities between decision trees were calculated using Equation (2), and the lower triangular similarity matrix *M*_*s*_ (Equation (3)) was formed.
(3)$$M_{S}=[\begin{array}{ccccc} 0 & 0 & \ldots & 0 & 0\\ M_{2,1} & 0 & \ldots & 0 & 0\\ \vdots & \vdots & & \vdots & \vdots \\ M_{q,1} & M_{q,2} & \ldots & M_{q,q-1} & 0 \end{array}]$$

The *M*_2,1_ calculated by Formula 2 is the similarity between tree 2 and tree 1. Then, the decision tree pair with the lowest similarity were regarded as a cluster, and the decision tree with the better classification accuracy in this cluster was chosen as the new decision tree. To avoid the decision trees decreasing too fast, the number of clusters *N*_*c*_ for evolution was set. By repeating the clustering evolution for *i* times, the random forest reached the highest prediction performance and the amount of the final decision trees was *n* − *i**N*_*c*_ (*i* = 1, 2, 3 ⋯ *n*). The prediction accuracy of decision tree was defined as
(4)$$Acc_{x}=N_{vx}/N_{v}$$
where *A**c**c*_*x*_ is the prediction accuracy of tree *x*, *N*_*v**x*_ is the number that predicted by tree *x* in *S*_*v*_ correctly, and *N*_*v*_ is the size of *S*_*v*_.

### 2.4. Parameter Optimization Adjustment

For the genetic clustering random forest, the combination performance of the initial decision tree size, the evolution times of genetic algorithm and clustering evolution were examined, and then the best parameter combination was selected.

Firstly, the size of initial decision trees, the evolution times of genetic algorithm and clustering evolution were defined in [*a*, *b*], [*c*, *d*] and [*e*, *f*]. Then, all the parameter combinations were evaluated. Thirdly, the steps above were repeated for *N**_adjust_* times to avoid the difference due to the initial data sets. Finally, an optimal combination was extracted for the genetic clustering random forest.

### 2.5. Important “Subregion-Gene Pairs” Determination

The *S**test* was used to test the prediction accuracy and the universality of the final random forest. Since the features in final decision trees distinguished AD and HC, it showed that the differences in characteristics between AD and HC were extremely significant. Therefore, these features were defined as important pairs. AD pathogenic genes and abnormal hippocampal subregions were further defined based on the important pairs. The important features were picked out for the following steps.

Firstly, the frequencies of features in the final decision trees were counted, and features were sorted by the frequency. Subsequently, the features were separated into several subsets, and these subsets were evaluated by a traditional random forest. Then, the subset with best classification capability was defined as the important “subregion-gene pair”. Finally, the frequency of subregions and genes in important pairs were counted. The top *N_f_* subregions and genes were considered as abnormal hippocampal subregions and AD pathogenic genes according to the frequency.

## 3. Results

### 3.1. The Results of Fusion Feature

According to Section 2.1 and Section 2.2, we calculated the correlations between hippocampal subregions and SNPs, such as heritability, regression coefficient and asymptotic *p*-value for t-statistic and extracted 123 genes with the SNPs number ≥ 200 in each gene. Then, the SNPs in each gene were separated into 10 groups equally. The corresponding correlation sequences were also separated in the same way. Finally, Pearson correlation coefficients of gene sequences and correlation sequences were calculated, and 1476 “subregion-gene pairs” were obtained from each group.

### 3.2. The Results of Parameter Optimization

Initially, $\sqrt{1476}$ ≈ 38 features were extracted from the original data set randomly as the elements to construct a decision tree. According to this step, a random forest with 300 decision trees were selected. Subsequently, the evolutionary times was set to 5, and the obtained random forest was used as the initial population for the genetic algorithm. After this, the similarities and differences between decision trees were further amplified, and a new random forest was constituted by these decision trees. Then, the clustering evolutionary with a step size of 10 was applied to the resulting random forest, and the evolution generations was 20. Based on the process above, we obtained a basic genetic clustering random forest.

To obtain the optimal parameter combination, the strategy described in Section 2.4 was used for the three parameters optimization. Firstly, the size of initial random forest, the evolution times of genetic algorithm and clustering evolution were in the interval of (300, 500), (1, 10) and (1, 20). Then, the classification performances of all parameter combinations were counted. Specifically, the size of the random forest started from 300 with a step size of 20 and ended at 500. Each different initial forest was genetically clustered in 200 parameter combinations to obtain the optimal genetic clustering combination. To avoid the difference due to the initial data sets, the steps above were repeated for 10 times and the optimal combination in each time was selected. The highest prediction performance in different initial forests and their corresponding genetic clustering parameter combinations are shown in Figure 5. We find that the peak value is at the node of the random forest size 480. The corresponding parameter combination is {3, 17}. Therefore, the best parameter combination with the optimal classification ability of the genetic clustering random forest is {480, 3, 17}.

### 3.3. Comparison with Other Methods

Besides the methods described in Section 3.2, the traditional random forest, the genetic algorithm random forest and the clustering evolutionary random forest were applied to select the optimal features.

Traditional Random Forest:

The numbers of decision trees in traditional random forest were also in (300, 500). To ensure that the results are credible, we used the same training set and validation set to optimize the model. The accuracies of the random forests and their size are shown in Figure 6, and the best size of the initial forest was 420.

Genetic Algorithm Random Forest:

To find the best genetic evolution times, the initial decision trees was evolved 500 times using the genetic algorithm. Then, the genetic algorithm random forest was constructed. Figure 7 displays the accuracies of the genetic algorithm random forest and the parameter combinations, and the best parameter combination is {500, 469}.

Clustering Evolutionary Random Forest:

The clustering evolutionary random forest was described in [14]. Compared with the genetic clustering random forest, the difference between them was whether there was a process of genetic evolution. Therefore, the size of initial random forest and the clustering evolution times were in the interval of (300, 500) and (1, 20). As shown in Figure 8, the prediction performance reached the peak with the size of 500 and evolution times of 18.

Comparison of the Four Methods:

We applied the test set *S**test* to evaluate the classification capability of the four methods, and the experiments were repeated 10 times with the selected parameter combination in each method. The accuracies and the corresponding number of experiments are displayed in Figure 9. As shown in Figure 9, the genetic clustering random forest model hade good prediction accuracy. In genetic clustering random forest and genetic algorithm random forest, the peaks of prediction accuracy exceeded 90%, while the peaks of the other two methods were all below 90%. The curve in Figure 9 also shows that the genetic clustering random forest had good stability. In 10 repeated experiments, the gap of the accuracy was less than 10%. These analyses proved the satisfied ability in classification and stability of the genetic clustering random forest.

### 3.4. The Extraction of Fusion Features

The analysis above proved that the features selected by the genetic clustering random forest had more effective classification. The essence of these features was the Pearson correlation between subregions and genes. Therefore, by analyzing the features in the final decision trees, important “subregion-gene pairs” could be identified.

The features in the final decision trees were resolved into “subregion-gene pairs”, and then the number of occurrences of each “subregion-gene pair” was counted. The top 500 pairs were candidate “subregion-gene pairs”. Table 2 lists the top 15 pairs with numbers greater than 20. However, only part of these candidate “subregion-gene pairs” had strong distinguishing ability. In order to define abnormal subregions and genes, it was necessary to extract the “subregion-gene pairs” with high contribution from these features. Firstly, the subsets size of candidate “subregion-gene pairs” was set in (70, 500), and the step size was 5. Then, a traditional random forest with 340 decision trees was used to test the classification ability. As displayed in Figure 10, the accuracy of the random forest reached the peak 83.3%. Therefore, the top 475 “subregion-gene pairs” were the important “subregion-gene pairs”. The top 475 “subregion-gene pairs” and the first 15 important “subregion-gene pairs” are shown in Figure 11. The details of top 475 important hippocampal subregions and genes are in Appendix A.

We defined the abnormal subregions and pathogenic genes according to the experiment results above. The subregions and genes with a high frequency were the abnormal subregions and pathogenic genes of hippocampus in AD.

Table 3 shows the important “subregion-gene pairs” that were found by four methods. The number of important features selected by the genetic clustering random forest was the least. Interestingly, although the genetic evolution was used in two methods, there were still the highest overlapping features ratio among the optimal features extracted by the two methods. Another interesting finding is that the method with a higher overlap ratio with our method had a higher classification ability (Figure 9). This proved that the classification performance of features in genetic clustering random forest was the highest and suggested that the process of genetic algorithm was significant to the classification.

In case of small sample size, the robustness and generalization of the proposed model need to be verified. Therefore, we conducted the following experiments. We constructed the fusion features based on two datasets (262HC+269EMCI and 262HC+288LMCI) and applied the genetic clustering random forest to calculate the parameter combinations and accuracies. To avoid the high accuracy occasional, the 12 independent experiments were performed, and the best and worse results were deleted. The information of datasets and parameter combinations are listed in Table 4, and the accuracies of 10 independent results are shown in Figure 12.

As shown in Table 4, the proposed model achieved satisfactory classification accuracy in different datasets by simply adjusting parameters. In addition, the curves of the three datasets classification accuracy in Figure 12 also proved the stability of the proposed model. The verified analysis proved that the feature construction method and the genetic clustering random forest had good applicability and classification ability.

## 4. Discussion

In this work, we proposed a method to construct the fusion features using multimodal data. Particularly, we proposed a genetic clustering random forest based on genetic algorithm for detecting fusion features constructed by subregions and genes.

Prior research on multimodal data focused on the structural covariance networks of white and gray matter [29,30,31]. These were applied to study the correlation between multimodal structural covariance networks and aging or aging-related pathologies [29,30], and suggested that these structural covariance networks had a good classification [32]. Another study applied the multimodal neuroimaging of structure and function to diagnose the Parkinson’s disease and HC [33]. Although these were multimodal data studies, they were all based on the fusion of the same data sources. An interesting and different method to construct the fusion features from multimodal data was described in [14]. Bi et al. fused the gene sequence data and time series of fMRI data to classify AD and HC. In this study, we proposed a novel method to construct fusion features, which had the following two benefits. Since there were differences between the MRI scans of AD and HC groups, we performed GWAS using the MRI data as phenotypes. The aim of applying GWAS was to obtain correlations, and the correlations between SNPs and phenotypes were usually used to identified significant SNPs. The use of GWAS enlarged them and found some significant SNPs. This was the first advantage. Since SNPs were in genes and had corresponding correlations, the significant correlations had corresponding genes. In addition, the significant SNPs and correlations of the AD group and HC group are quite different. Using the characteristics of these genes and correlations, the differences of fusion features between the AD group and HC group were further amplified. This was another advantage. Therefore, compared to the method in [14], we used correlation sequences instead of image sequences to construct the fusion features.

For the feature’s detection, the genetic clustering random forest based on the genetic algorithm was proposed as a novel and improved method. Compared to the method in [14], we applied a genetic algorithm before clustering evolution. The genetic process drew on the idea of clustering evolution to select parents with high classification accuracy, and the similarities between the generated decision trees were low. The advantage of this was that decision trees with high classification accuracy were retained. As shown in Figure 9, the classification accuracy of genetic clustering random forest is the best of the four methods. Additionally, the accuracy of genetic algorithm random forest is also better than the other two. The parent selection strategy in genetic clustering random forest and genetic algorithm random forest draws on the idea of clustering evolution and parents are selected based on the similarity between decision trees. These shows that the combination of genetic algorithm and clustering evolution has an effective grouping effect in the evolution of random forest. In traditional classification methods [34,35], a single learner is common. In the improved methods [14,36,37] based on a learner, the ensemble learning is used to enhance the classification performances of the models. In our proposed model, the idea of a genetic algorithm is introduced to evolve the initial decision trees. The diversities of decision trees in the same group were further reduced and the differences between AD and HC were enhanced. Although the accuracies of four methods in validation set were similar (Figure 5, Figure 6, Figure 7 and Figure 8), the accuracy of genetic clustering random forest in the test set was obviously higher than in the other three methods (Figure 9). Additionally, we observe that the stability of genetic clustering random forest was better than others (Figure 9). We can also observe that the model had good generalization performance in different datasets in Table 4 and Figure 12. These demonstrate that the genetic clustering random forest had good predictive classification ability and generalization.

The “subregion-gene pairs” that classify AD and HC well may be the potential pathogenic factors of AD. Some abnormal subregions and pathogenic genes associated with AD were detected in our research, such as hippocampus amygdala transition area (HATA), fimbria, parasubiculum, hippocampal fissure and *RYR3* and *PRKCE*. The HATA was connected with the amygdala closely, and compared with the healthy group, the volumes of HATA were reduced in the MCI group [2,38]. In another study, obvious changes in fimbria were observed in AD [39]. The change of parasubiculum affected the medial temporal memory system and dementia, and AD patients had more cellular neurofibrillary tangles in parasubiculum [40,41,42,43]. 

We counted the overlaps of the genes identified in our study and the genes of the top 26 “important brain region-gene pairs” in [14]. Only *KAZN* and *RF00019* were not included in our study. This demonstrated that most of the same genes were obtained using different methods and data sets. However, we found the overlaps of fusion features were 73 using the two methods in our data set (Table 3). The randomness of the genetic algorithm is the main reason. These 73 features had a great contribution to classification, and the classification accuracy and identified features of our method are higher than those in [14]. It can be inferred that these more identified features improve the classification accuracy, and the genes in these features can be speculated as AD candidate genes. Among these genes, priori research showed that *RYR3* identified the association with AD using multifactor dimensionality reduction [44]. The upregulated level of RYR3 and a significant interaction between *RYR3* and *CACNA1C* were observed in the AD group [45,46]. Gong et al. found four disease related SNPs (rs965471, rs10519874, rs7498093 and rs17236525), and proved that RYR3 had shared genetic susceptibility in hypertension, diabetes, and AD [47]. According to our founding, *PRKCE* detected by our method tend be associated with AD. The previous study has proved that the endothelin-converting enzyme activity increased by overexpression of *PRKCE* reduces the α*A*β levels [48]. Based on the above analysis, the part of the abnormal subregions and pathogenic genes identified are related to AD. Therefore, the remaining genes can be speculated as AD candidate genes. The discovery of these subregions and genes by our method provides new candidate genes for the future research of AD and is significant to the study of the potential mechanism in the hippocampus.

## 5. Conclusions

The genetic clustering random forest proposed in this paper provides a novel method for detecting the abnormal “subregion-gene pairs” in the hippocampus. This method constructs decision trees through random forest, evolves the decision trees genetically through genetic algorithm and performs cluster evolution on the results obtained. Finally, the important “subregion-gene pairs” were extracts based on the fusion features that were constructed by subregions and genes. Furthermore, we also show that our method had higher accuracy than the traditional random forest, the genetic algorithm random forest and the clustering evolutionary random forest.

In this paper, the study of detecting abnormal subregions and genes using genetic clustering random forest had the following strengths. (1) We improved a more efficient method to construct the fusion features. This method reduced the differences between the subjects in the same group and increases the differences between AD and HC groups. (2) We improved a genetic clustering random forest based on the genetic algorithm to detect the features. The evolution of training set using genetic algorithm amplified the differences between decision trees too. (3) We also show that the classification ability and stability of our method were better than other conventional methods.

Since AD also has other markers, in the future, we will continue to look for fusing other data such as protein and RNA to construct the fusion features. Further research needs to be carried out to verify the correlations between candidate genes and AD.

## Figures and Tables

**Figure 1 genes-12-00683-f001:**
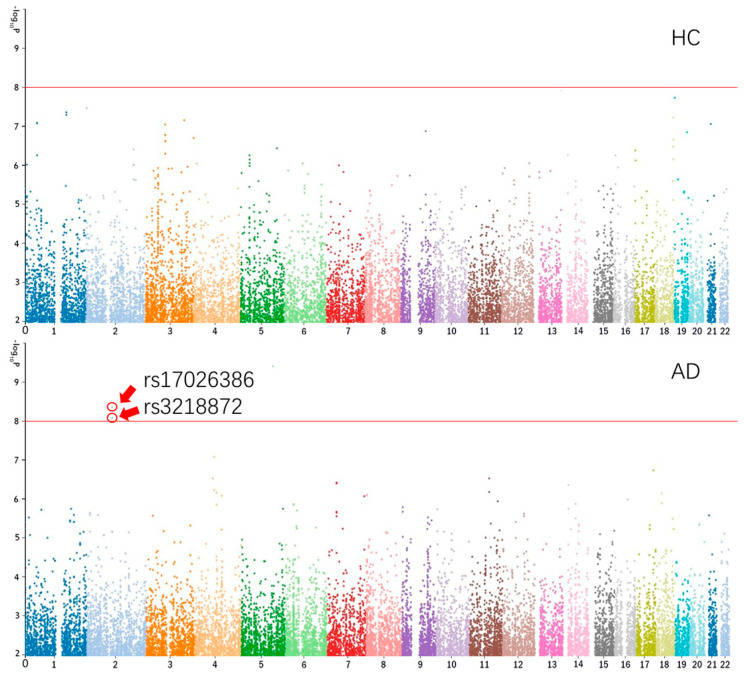
The Manhattan plots of CA1 of HC and AD. CA1 = cornu ammonis 1 region; HC = healthy control; AD = Alzheimer’s disease.

**Figure 2 genes-12-00683-f002:**
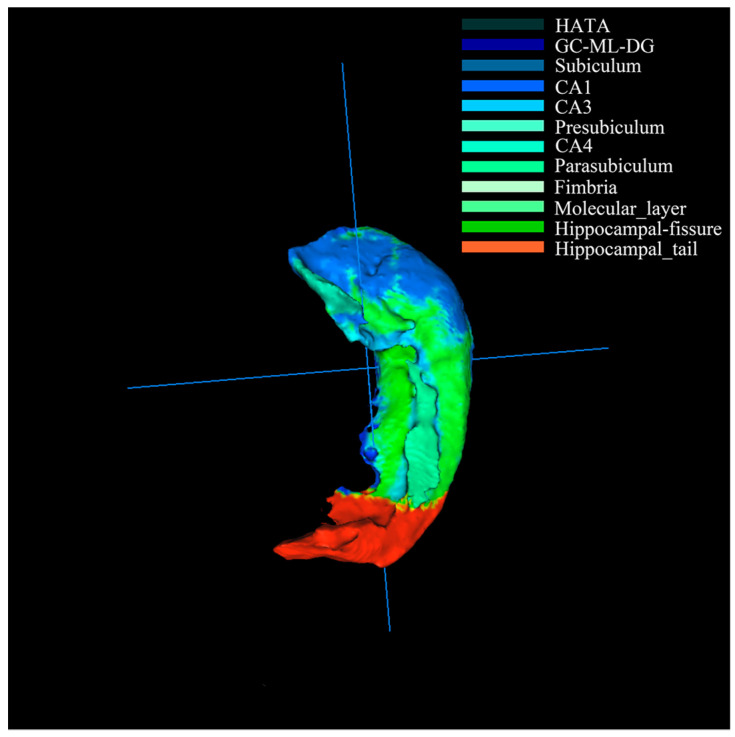
The anatomical representation of the 12 hippocampal subregions. The color represented different subregions. HATA = the hippocampus amygdala transition area; GL_ML_DG = the granule cell layer and molecular of the dentate gyrus; CA1 = cornu ammonis 1 region; CA3 = cornu ammonis 3 region; CA4 = cornu ammonis 4 region.

**Figure 3 genes-12-00683-f003:**
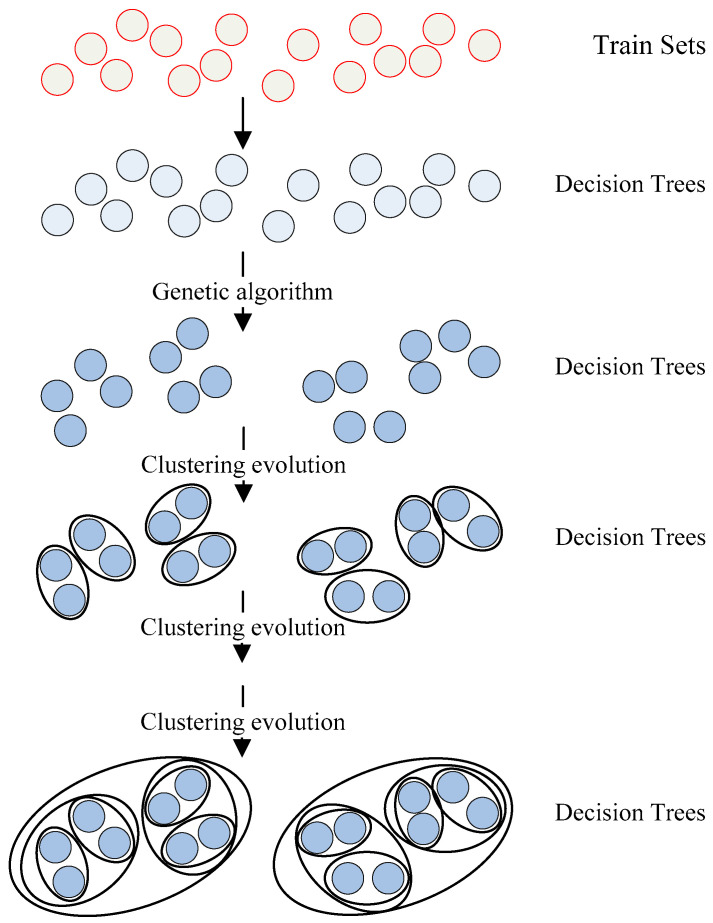
The schematic diagram of genetic clustering random forest. Genetic algorithm and clustering evolution were applied to increase the difference among basic classifiers and further improve their diversity and accuracy.

**Figure 4 genes-12-00683-f004:**
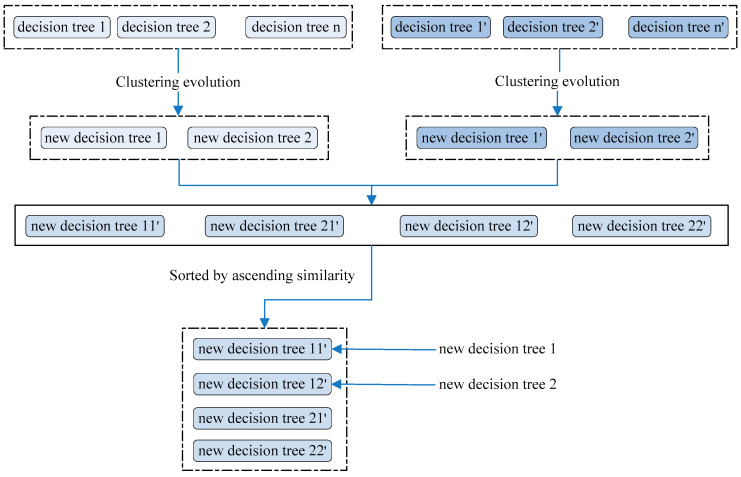
The schematic diagram of genetic evolution. The clustering evolutionary was used to select the parent.

**Figure 5 genes-12-00683-f005:**
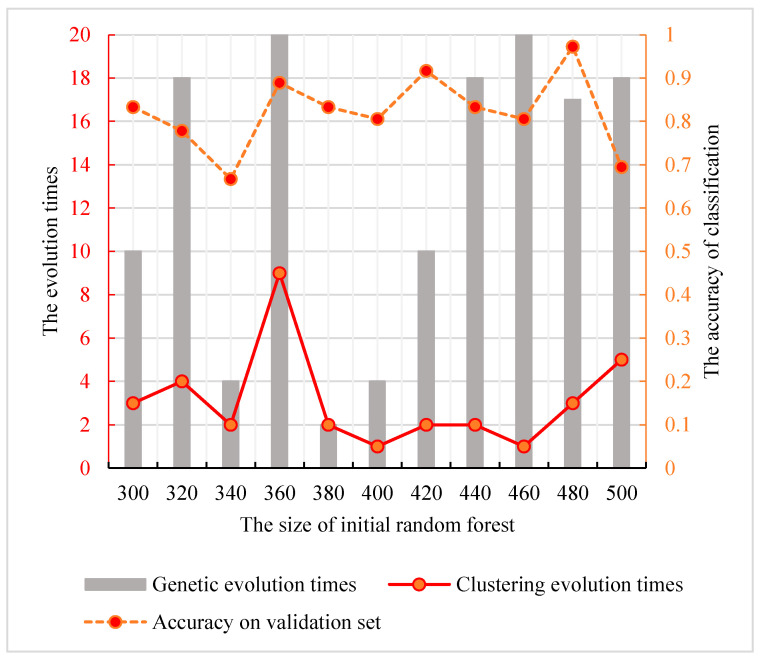
The relationship among the clustering evolution times, the genetic evolution times and the size of initial random forest in genetic clustering random forest. The dotted line indicates the accuracy of classification. The solid lines and bars indicate the number of genetic evolution times and clustering evolution times according to the initial random forest size.

**Figure 6 genes-12-00683-f006:**
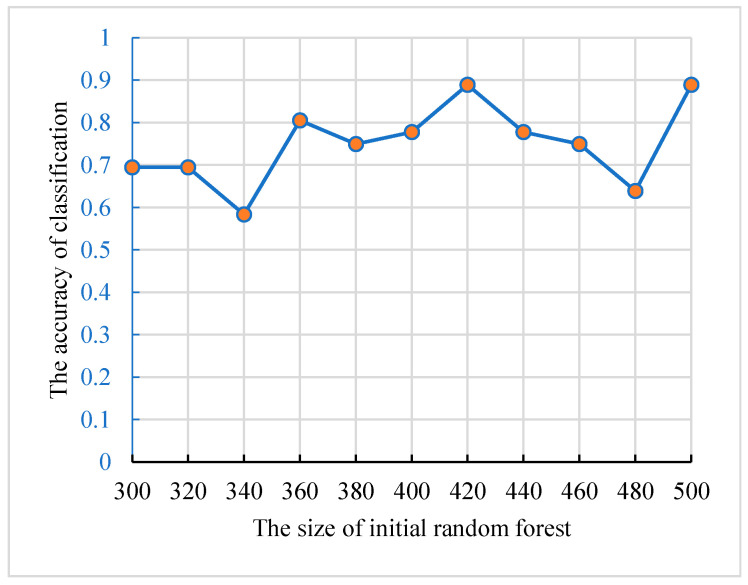
The relationship between the accuracies and the size of initial random forest.

**Figure 7 genes-12-00683-f007:**
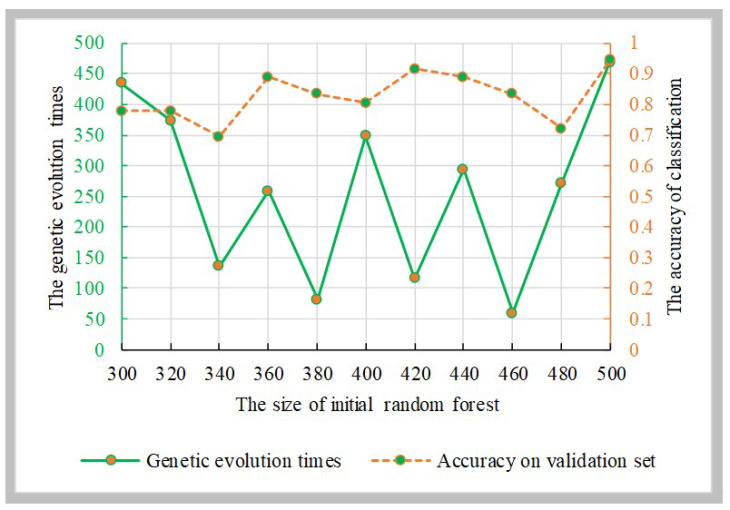
The relationship between the genetic evolution times and the size of initial random forest.

**Figure 8 genes-12-00683-f008:**
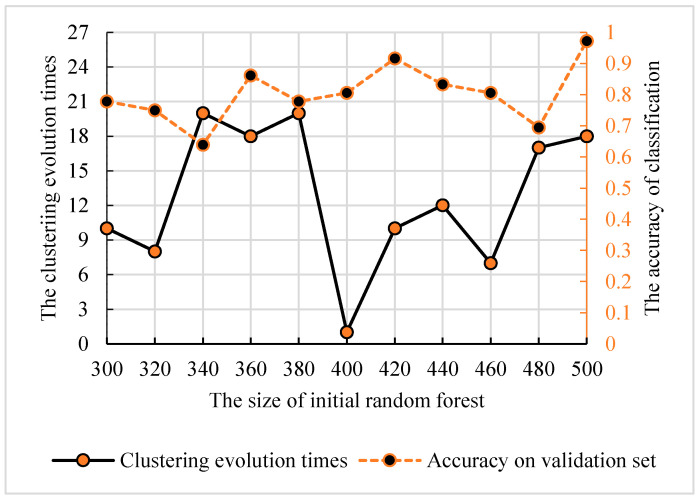
The relationship between the clustering evolution times and the size of initial random forest. The dotted line indicates the accuracy of classification. The solid lines indicate the number of clustering evolution times according to the initial random forest size.

**Figure 9 genes-12-00683-f009:**
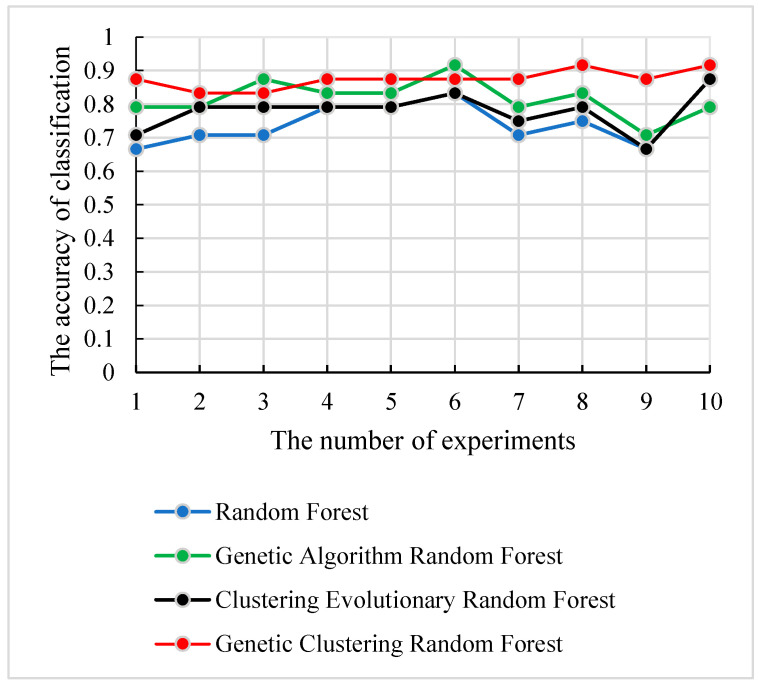
The relationship curves of accuracy and the four methods in 10 experiments.

**Figure 10 genes-12-00683-f010:**
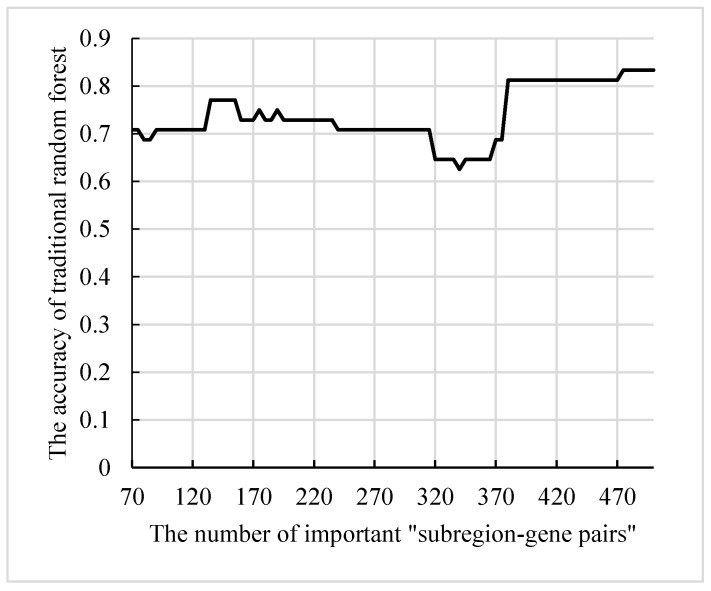
The ability of the traditional random forest to classify the subsets.

**Figure 11 genes-12-00683-f011:**
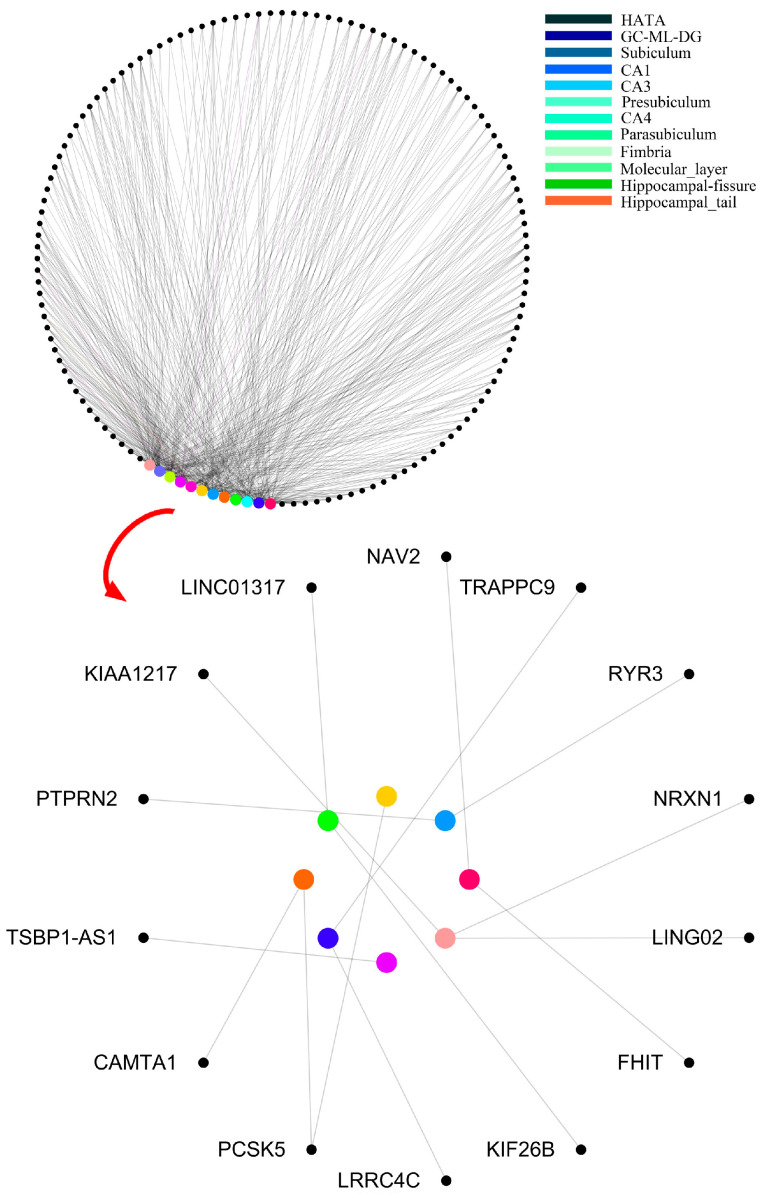
The top 475 “subregion-gene pairs” and the first 15 important “subregion-gene pairs”. Nodes denote the subregions and genes. Edges denote the association between subregions and genes, and the widths of edges denote the frequency of each “subregion-gene pair”.

**Figure 12 genes-12-00683-f012:**
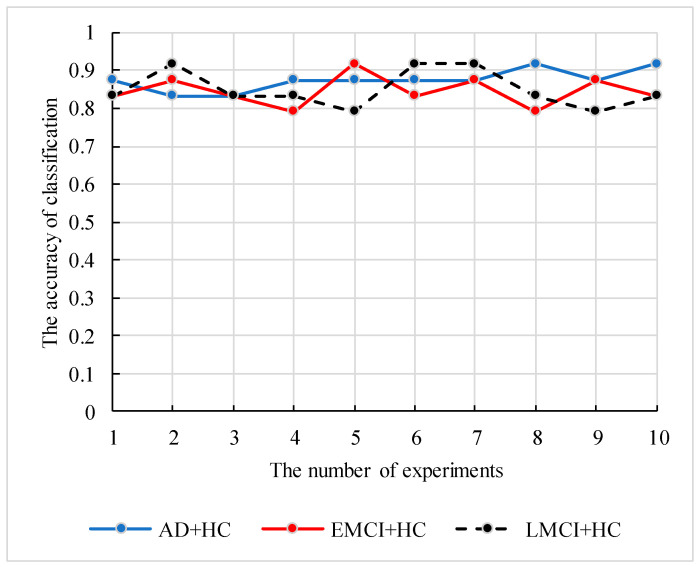
The classification accuracy curve of the proposed model based on three datasets.

**Table 1 genes-12-00683-t001:** Participant characteristics. HC = healthy control; AD = Alzheimer’s disease; M/F = male/female; Edu = education; sd = standard deviation.

Subjects	HC	AD
Number	262	125
Gender (M/F)	135/127	76/49
Age (mean ± sd)	74.6 ± 5.8	74.3 ± 7.7
Edu (mean ± sd)	16.4 ± 2.8	15.8 ± 3.0

**Table 2 genes-12-00683-t002:** The important “subregion-gene pairs” with numbers greater than 20. GL_ML_DG = the granule cell layer and molecular of the dentate gyrus; CA4 = cornu ammonis 4 region.

Numbers	Subregions	Genes
29	PARASUBICULUM	CAMTA1
25	PARASUBICULUM	PCSK5
23	HIPPOCAMPAL_FISSURE	TSBP1-AS1
23	FIMBRIA	LRRC4C
22	GL_ML_DG	KIF26B
22	CA4	LINGO2
22	CA4	NRXN1
22	FIMBRIA	TRAPPC9
21	MOLECULAR_LAYER	FHIT
21	MOLECULAR_LAYER	NAV2
21	GL_ML_DG	LINC01317
21	CA4	KIAA1217
21	PRESUBICULUM	PCSK5
21	CA3	PTPRN2
21	CA3	RYR3

**Table 3 genes-12-00683-t003:** The important “subregion-gene pairs” identified by the traditional random forest. GCRF = genetic clustering random forest; RF = random forest; GARF = genetic algorithm random forest; CERF = clustering evolutionary random forest.

Method	Discoveries	Overlap with Our Method
GCRF	475	-
RF	205	68
GARF	90	35
CERF	220	73

**Table 4 genes-12-00683-t004:** Model validation experiments on different datasets. GE = genetic evolutionary times; CE = clustering evolutionary times; HC = healthy control; EMCI = early mild cognitive complaint; LMCI = late mild cognitive complaint; AD = Alzheimer’s disease.

Dataset	Base Classifier Number	GE Times	CE Times	Optimal Features Number	Average Accuracy
125AD + 262HC	480	3	17	475	87.50%
269EMCI + 262HC	460	1	3	165	84.58%
288LMCI + 262HC	400	5	14	470	85.00%

## Data Availability

The data is available at http://adni.loni.usc.edu/.

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
