# Peer review of "Hippocampal Subregion and Gene Detection in Alzheimer’s Disease Based on Genetic Clustering Random Forest"

_genes, 2021, doi:10.3390/genes12050683_

Round 1
Reviewer 1 Report
I would like to thank the Authors and Editors for the opportunity to review this manuscript, which I find extremely interesting in introducing a new computational method for the discovery of gene-brain area associations in Alzheimer’s disease.
I find the work of notable importance and quite clearly explained in all its sections, although I have some comments for the Methods section.
Here are my comments:
- can you specify if all the patients were explicitly diagnosed with AD, and how was this carried out? (e.g. molecular analysis, imaging…)
- can you briefly specify the criteria for distinguishing early and late MCI in the text?
- line 47: you have not defined what "DG" stands for
- lines 61-62: although intuitive, please define "EMCI" and "LMCI"
- lines 116-120: methodology is unclear: how did you map the SNPs into their genes? What reference human genome version did you use? How did you select the top N genes? The criterion is not clear until ones reads the subsequent paragraph.
- all the acronyms used in the tables should be described explicitly in a subtext attached to the table, or in the main text.
Thank you and good work!
Reviewer 2 Report
Li and colleagues describe their computational approach for a more detailed functional partitioning of the hippocampus concerning AD. My comments are listed below in the manner they first occur to me perusing the manuscript.
- Line 102. Please briefly describe these processes here.
- Line 104. What platform array were the SNPs generated on? It seems that no effort to input additional SNPs was made? Why not?
- Line 113. This is quite vague! Explain how "gene sequences" are defines.
- Line 116. An anatomical representation of the hippocampus with the different regions highlighted should be provided here.
- Line 117. What statistical approach was used to perform the GWAS, and what software?
- Line 119-120. How many SNPs were mapped onto genes? What construes a top Ngene?
- Line 122. Explain what a correlation sequence is? How were the biological and technical covariates handled in the Pearson correlations?
- Line 151. How were similarities between trees calculated? Below the authors talk about Euclidean distance; is that how they calculated similarities?
- Line 205. Specify the exact number here! What is the density distribution for these SNPs across the 123 genes? On average, you have 2 SNPs per gene. How reliable is your test statistics with only 2 SNPs per gene, if my assumption is correct?
- Line 205. What cut off p-value was used here? I presume these were not corrected for multiple testing? If so, how was the multiple test issue adjusted for?
Round 2
Reviewer 1 Report
Thank you for the opportunity to comment on the revised version of this Manuscript, which I find has been improved by the Authors. The Reviewers' comments have been adequately addressed and integrated. I find this an excellent and innovative work, without further need for corrections on my part.
Thank you for your work.
Reviewer 2 Report
While the authors have responded to the reviewers' comments, their responses, unfortunately, did not clarify but further complicated what exactly did they do. My comments are provided below.
- Line 133: So the authors performed a GWAS between 564K SNPs and their imaging data via linear regression analysis, adjusting for the biological covariates. Where's the Manhattan plot of these results. Were, these corrected for any multiple tests. Were the data adjusted for population stratification?
2. Line 122 is utterly confusing! So genes were defined as a number sequence. Did the authors take into consideration any potential LD relationship between the SNPs? First I don't see the benefit of randomly assigning numbers to SNPs within a gene if that what they did. It's much better for them to assess the LD between SNPs and simply use the haplotype structure for their correlation analyses. Additionally, I'm unsure how is the gene still defined here. As SNPs are bi-allelic, it will matter which base is used for the corresponding gene number sequence.
3. Line 139: So how many genes these SNPs mapped onto? After all the authors are interested in assessing genes, not random sequences that may contain an SNP in the middle of nowhere. I'm still confused about what Nsnp means?
4. Finally in response 10, their response is a circular statement that is also absolutely wrong, questing the validity of their entire manuscript.
